# A Zn(II)–Metal–Organic Framework Based on 4-(4-Carboxy phenoxy) Phthalate Acid as Luminescent Sensor for Detection of Acetone and Tetracycline

**DOI:** 10.3390/molecules28030999

**Published:** 2023-01-19

**Authors:** Nairong Wang, Shanshan Li, Zhenhua Li, Yuanyuan Gong, Xia Li

**Affiliations:** Department of Chemistry, Capital Normal University, Beijing 100048, China

**Keywords:** luminescent sensor, tetracycline, acetone, metal–organic framework

## Abstract

As hazardous environmental pollutants, residual tetracycline (TC) and acetone are harmful to the ecosystem. Therefore, it is necessary to detect the presence of these pollutants in the environment. In this work, using Zn (II) salt, 4-(4-carboxy phenoxy) phthalic acid (H_3_L), and 3,5-bis(1-imidazolyl) pyridine (BMP), a new metal–organic framework (Zn-MOF) known as [Zn_3_(BMP)_2_L_2_(H_2_O)_4_]·2H_2_O was synthesized using a one-pot hydrothermal method. The Zn-MOF has a three-dimensional framework based on the [Zn1N_2_O_2_] and [Zn2N_2_O_4_] nodes linked by a tridentate bridge BMP ligand and an L ligand with the μ_1_:η^1^η^0^/μ_1_:η^1^η^0^/μ_0_:η^0^η^0^ coordination mode. There were two kinds of left- and right-handed helix chains, Zn1-BMP and Zn1-BMP-Zn1-L. The complex was stable in aqueous solutions with pH values of 4–10. The Zn-MOF exhibited a strong emission band centered at 385 nm owing to the π*→π electron transition of the ligand. It showed high luminescence in some common organic solvents as well as in the aqueous solutions of pH 4–10. Interestingly, TC and acetone effectively quenched the luminescence of the Zn-MOF in aqueous solution and enabled the Zn-MOF to be used as a sensor to detect TC and acetone. The detection limits of TC and acetone were observed to be 3.34 µM and 0.1597%, respectively. Even in acidic (pH = 4) and alkaline (pH = 10) conditions, the Zn-MOF showed a stable luminescence sensing capability to detect TC. Luminescence sensing of the Zn-MOF for TC in urine and aquaculture wastewater systems was not affected by the interfering agent. Furthermore, the mechanism of sensing TC was investigated in this study. Fluorescence resonance energy transfer and photoinduced electron transfer were found to be the possible quenching mechanisms via UV–Vis absorption spectra/the excitation spectra measurements and DFT calculations.

## 1. Introduction

Environmental pollutants easily spread into the soil, air, and water environments due to their high solubility and mobility. The high prevalence of these pollutants in the ecosystems is gravely threatening the environment and human health and has become a global concern [1]. The massive use of antibiotics and pesticides has led to the emergence of super-resistant bacteria [2]. Residual solvent molecules in the environment can also cause cancer, malformation, neurotoxicity, and other health problems [3]. Belonging to the class of broad-spectrum antibiotics, tetracycline (TC) is widely used in treating human and animal bacterial infections due to its low toxicity, low cost, and excellent oral absorption. However, TC is not fully absorbed during animal metabolism and enters the environment with feces. Hence, large amounts of TC residues are often found in soil and water environments [4,5,6]. In addition, acetone is a typical volatile organic solvent widely used in cosmetics, adhesives, and other commercial products. Long-term exposure to acetone may damage the liver, kidneys, and nerves, causing inflammation [7,8,9]. Acetone is readily found in industrial wastewater environments. Currently, high-performance liquid chromatography (HPLC), capillary electrophoresis (CE), and immunoassay are common methods for detecting environmental pollutants such as tetracycline and acetone [10,11,12]. However, these methods generally require expensive and sophisticated instruments, complex pre-processing procedures, and skilled technicians, which are not conducive to rapid and routine monitoring [13,14,15,16]. Therefore, it is crucial to develop a fast and simple method for detecting TC and acetone.

Metal–organic frameworks (MOFs), a group of porous materials composed of inorganic metal ions and organic bridging ligands, have sparked a lot of academic interest in recent years [17,18,19]. MOFs have flexible structural designs, highlighting a specific surface area and highly ordered pores. Therefore, MOFs have been widely explored for the storage and separation of various gases as well as catalytic [20,21,22,23], biomedical [24], magnetic [25,26], and chemical sensing applications [27,28,29]. Luminescent metal–organic frameworks, as an important family of MOF materials, have made outstanding contributions in detecting harmful pollutants such as antibiotics, solvent molecules, pesticides, explosives, and other pollutants due to their superior luminescence ability [30,31,32,33,34]. In recent years, MOFs have also made positive advances in detecting tetracycline and acetone. Huang et al. synthesized an Fe-modified MXene-derived MOF that can be used as a high-performance acetone sensor [35]. Gan et al. proposed fluorescent Eu-MOF designed to detect TC rapidly in food samples [36]. Zn(Ⅱ) ion with d^10^ electronic configuration possesses not only various coordination modes but also attractive luminescence properties when bound to functional ligands. Due to these characteristics, the Zn-MOFs have been reported as sensors for detecting antibiotics, metal ions, and small molecules [37,38,39]. For example, {[Zn(L)_0.5_(bpea)]·0.5H_2_O·0.5DMF}n has been reported to be an effective luminescence sensor for nitrofurazone (NFT) in aqueous solutions with an LOD of 0.35 µM [40]. [Zn_2_(tdca)_2_(bppd)_2_]·2DMF can be used as a sensitive luminescence probe for the detection of Cd^2+^ [41].

In general, the preparation of luminescent MOFs requires the selection of ligands with luminescence properties and functional groups with the ability to modify the luminescence properties, in addition to different photoactive metal ions [15]. The luminescence properties of MOFs can be adjusted by introducing organic ligands with aromatic groups or conjugated π-systems through conjugation effects [42]. Due to their strong coordination, aromatic carboxylic acid ligands are the best candidates to form MOFs. 4-(4-carboxy phenoxy) phthalic acid (H_3_L) is a V-type semi-rigid carboxylic acid. The two benzene rings of the ligands can rotate through ether groups and exhibit some conformational flexibility [43]. In addition, it contains three carboxylic acid groups, with the metal center in different ligand coordination ways. It makes the formation of MOFs easy to obtain and with great structural diversity. It has been reported that a series of Zn(Ⅱ) complexes constructed by the H_3_L ligand have 3D frameworks and good luminescence properties, which provide the possibility for the synthesis of the luminescent Zn-MOF [44]. The introduction of N-containing heterocyclic organic ligands can enrich the design of the complex. 3,5-bis(1-imidazolyl) pyridine (BMP) is a rigid nitrogenous heterocyclic ligand with large conjugated groups that readily form coordination bonds with metals and increase the stability of the structure [45,46].

Here, in this work, the H_3_L ligand and BMP ligand were used to synthesize new MOFs by employing the hydrothermal method: [Zn_3_(BMP)_2_L_2_(H_2_O)_4_]·2H_2_O. The Zn-MOF is a three-dimensional microporous framework and showed strong and stable luminescence in a solid-state and aqueous solution. The luminescence properties of the Zn-MOF in common solvents and aqueous solutions with different pH values were investigated. The application of the Zn-MOF as a sensor in detecting tetracycline and acetone was also explored in this study (Figure 1).

## 2. Results and Discussion

### 2.1. The Crystal Structure of [Zn_3_(BMP)_2_L_2_(H_2_O)_4_]·2H_2_O

The Zn-MOF featured a 3D structure, which was crystallized in a monoclinic system with a P2_1_/n space group. The asymmetric unit of the Zn-MOF consists of three Zn(Ⅱ) ions, two L ligands, two BMP ligands, four coordinated water molecules, and two free water molecules. The Zn(II) ions have two different coordination modes. The Zn1 ions had a total of four coordinations: two coordinations with two O atoms from two L ligands and two coordinations with N atoms from BMP ligands, to form a slightly distorted tetrahedron geometry [Zn1N_2_O_2_] (Figure 1a). The bond length range of Zn1-O is 1.933(2)–1.946(2) Å and the bond length range of Zn1-N is 2.010(3)–2.024(2) Å. The Zn2 ions had six coordinations in the [Zn2N_2_O_4_] octahedral configuration, which were formed by linking to four O atoms from four water molecules and two nitrogen atoms from two BMP ligands. The Zn2-O bond distances vary from 2.079(2) to 2.118(2) Å and the Zn2-N bond length is 2.140(2) Å. Each BMP is a three-coordination ligand connected with three different Zn(Ⅱ) ions in μ_3_:η^1^η^1^η^1^ coordination mode to form a two-dimensional Zn-BMP network. The L ligand acted as a bridging ligand to link two adjacent Zn1 ions through the μ_1_:η^1^η^0^/μ_1_:η^1^η^0^/μ_0_:η^0^η^0^ coordination mode along the a-axis extension, and the Zn-BMP network further formed a three-dimensional structure (Figure 1b). Notably, the three-dimensional framework consisted of two types of helical chains (left-handed and right-handed) having a repeating unit [Zn1-BMP] which consisted of two BMP ligands and two Zn1 ions with a pitch of 15.8197 Å (Figure 1c). [Zn1-BMP-Zn1-L] was composed of a BMP ligand, an L ligand, and two Zn1 ions with a pitch of 15.6548 Å (Figure 1d).

### 2.2. Morphological Properties and Thermal Stability of the Zn-MOF

The morphology of the Zn-MOF was described using optical microscopy and SEM. The Zn-MOF was colorless and had transparent blocky crystals and featured a 3D structure with a rough surface (Appendix A). The Zn-MOF’s thermal stability was investigated using TGA on samples by raising the temperature from 25 to 800 °C at a rate of 10 °C/min. The TGA data indicated that the Zn-MOF exhibited good thermal stability (Appendix A). As the temperature increased, the complex began to lose weight continuously. The Zn-MOF complex showed a mass loss of 8.12% from 70 °C to 235 ℃, which was caused by the loss of free water molecules and coordinated water molecules (calculated value: 8.15%). When the temperature reached 280 °C, a sudden weight loss occurred, and the organic ligands in the Zn-MOF began to decompose. No further weight loss was observed after 510 °C. A total loss of 77.88% was observed until the temperature reached 510 °C when the decomposition of the Zn-MOF was completed (calculated value: 81.55%). The residual weight corresponded to the formation of ZnO.

### 2.3. The Solid-State Photoluminescence of the Zn-MOF

The solid-state photoluminescence of the H_3_L, BMP, and Zn-MOF were examined at room temperature (Figure 2). The spectrum shows that H_3_L and BMP ligands exhibited a maximum emission wavelength at 408nm (λ_ex_ = 358 nm) and 389 nm (λ_ex_ = 323 nm), respectively, while the Zn-MOF exhibited a broad emission band with a maximum emission wavelength of 385nm (λ_ex_ = 305 nm), which might be explained by the ligand’s π* → π electron transition. 

### 2.4. Luminescence of the Zn-MOF in Water with Different pH

One of the essential considerations for sensing applications is the stability of MOFs in aquatic environments. The Zn-MOF sample was immersed in water and aqueous solutions with pH of 4, 6, 9, 11, and 12 for 72 h. Comparing the PXRD patterns of the Zn-MOF sample in various aqueous solutions to the pattern obtained for the Zn-MOF crystal, there were no differences (Figure 3a). This verified that the structure of the Zn-MOF was unaltered. This result demonstrated the high chemical stability of the produced Zn-MOF in aqueous solutions over a broad pH range. Meanwhile, Zn-MOF was also the subject of a luminescence experiment in water at pH 1–14. The Zn-MOF powder (3 mg) was dispersed in aqueous solutions (3 mL) with various pH values, and the light-emitting properties were measured. As shown in Figure 3b, under strongly acidic conditions (pH < 3), the H_3_L ligand may not have coordinated well with metal Zn^2+^ ions, which resulted in different degrees of a decrease in fluorescence intensity. After the pH value of the solution exceeded 12 (pH > 12), the luminescence intensity of the Zn-MOF diminished sharply. However, at the emission wavelength of 385 nm, the luminescence intensity of the Zn-MOF remained nearly constant for pH values between 4 and 12. In addition, compared with the solid emission spectrum of the Zn-MOF, the emission band in aqueous solution was also observed to have the maximum value at 385 nm with relatively high intensity (Appendix A). This indicated that the prepared Zn-MOF maintained good luminescence properties in water. The stable luminescence properties suggested that the Zn-MOF could be a regular and potentially helpful material as a luminescent sensor in aqueous solutions. 

### 2.5. The Luminescence Properties of the Zn-MOF in Solvents and Sensing for Acetone

The luminescence of the Zn-MOF was investigated in the presence of common organic solvents to explore the stability of the Zn-MOF in a solvent environment. The Zn-MOF (3 mg) was dispersed in 3 mL of common solvents (water, acetone, dichloromethane, methanol, ethanol, DMA, DMF, DMSO, acetonitrile, etc.) to form suspensions. The luminescence spectra of these suspensions were obtained at an excitation of 305 nm. As shown in Figure 4, the Zn-MOF showed excellent luminescence performance in some commonly used organic solvents. However, it is worth noting that the luminescence of the Zn-MOF in acetone nearly disappeared, suggesting that the Zn-MOF can be used as a luminescent sensor to detect acetone molecules.

To further explore the influence of the acetone molecule on the luminescence intensity of the Zn-MOF, acetone solution was added to the suspension to perform luminescence titration experiments. Acetone solution in the quantity of 10 μL was added dropwise to a beaker containing a 3 mg sample, and the luminescence intensity of the system at 385 nm was monitored. As shown in Figure 5, when the volume of the acetone solution increased from 0 μL to 300 μL, the luminescence intensity of the Zn-MOF gradually decreased. When 300 μL of acetone solution was added, the luminescence quenching efficiency (QE) of the Zn-MOF was observed to be 67%. The luminescence intensity of the Zn-MOF had a specific linear relationship with the volume fraction of acetone added as per the following linear equation: I_0_/I−1 = K_SV_V.

Here, I_0_ and I are the luminescence intensity before and after adding acetone, respectively; V is the volume fraction of acetone in water; and K_SV_ is the slope of the linear equation [9]. The correlation coefficient (R^2^) was found to be 0.9907. The limit of detection (LOD) of the Zn-MOF for acetone was calculated as 0.1597%, indicating that the Zn-MOF can detect acetone molecules in a low concentration range.

To explore the selective detection of the Zn-MOF for the acetone molecule, the Zn-MOF sample (3 mg) was immersed in different organic solvents (ethyl acetate, ethanol, methanol, methyl ethyl ketone, dichloromethane, etc.). An equal amount of acetone was then added for the anti-interference experiment. As shown in Appendix A, it was found that only the presence of acetone molecules in organic solvents weakened the luminescence of the Zn-MOF, and the quenching effect of acetone on the luminescence of the Zn-MOF was not affected by the interfering solvents.

### 2.6. Luminescence Sensing of the Zn-MOF to TC

Residual TC has been identified as a major organic contaminant in water. Being present in aquatic ecosystems, this non-degradable contaminant is harmful to the environment and human health. Therefore, the creation of reliable and efficient techniques to identify TC pollutants in water is urgently needed. So, the luminescence sensing of the Zn-MOF to TC was investigated. The Zn-MOF samples (3 mg) were soaked in different antibiotics (1 × 10^−3^ mol/L, 3 mL), such as penicillin (PEN), chloramphenicol (THI), griseofulvin (GRI), erythromycin (ERY), streptomycin (STR), and kanamycin (KANA). The solutions were then subjected to ultrasound treatment for 30 min to obtain a uniform suspension. Figure 6a shows that TC weakened the luminescence of the Zn-MOF when different antibiotics were added to the suspension of the Zn-MOF. Moreover, upon introducing TC to the mixture of the Zn-MOF and the alternative antibiotic, the luminescent emission of the Zn-MOF was significantly quenched (Figure 6b). Therefore, the prepared Zn-MOF showed good selectivity and sensitivity to the detection of TC in the presence of other antibiotics.

To explore the effect of TC on the luminescence intensity of the Zn-MOF, a luminescence titration experiment was performed by gradually adding TC to a suspension of the Zn-MOF and monitoring the emission intensity at 385 nm (Figure 6c). When the volume of TC solution (2 × 10^−4^ mol/L) was increased from 0 to 220 μL, the emission intensity of the Zn-MOF gradually decreased. A good linear relationship between luminescence intensity and TC concentration was visible at low concentrations (Figure 6d). The calculated KSV was 1.97 × 10^5^ M^−1^, and the R^2^ was 0.9918. In addition, the LOD for TC was estimated to be 3.34 µM. In addition, taking pH = 4 or 10 as an example, the fluorescence sensing ability of the Zn-MOF to TC in acidic and alkaline conditions was also tested. The LOD of TC in slightly acidic and alkaline aqueous solutions (pH = 4 and 10) was calculated using fluorescence titration and was found to be 5.17 µM and 7.27 µM, respectively (Figure 7). It indicated that the Zn-MOF had the ability to detect TC in the aqueous system of pH 4–10. These results demonstrated the high selectivity and anti-interference capability of the prepared Zn-MOF for TC detection.

The most crucial aspects when evaluating a sensor material for practical applications are response time and recyclability. Measurements of the time-dependent luminescence were carried out to confirm the response rate of the Zn-MOF. The emission of the Zn-MOF was immediately quenched by exposure to TC for 10.11s (Appendix A), indicating rapid detection of TC with the luminescence of the Zn-MOF. Moreover, the recovery performance of the Zn-MOF as a TC luminescent sensor was evaluated to fulfil the recyclability requirements for potential practical applications. Therefore, a recycling experiment was conducted to assess the recycling and regeneration capabilities of the Zn-MOF. Firstly, the suspension containing TC was centrifuged and dried to recover the Zn-MOF powder, which was then washed with ethanol to remove the TC. The luminescence of the Zn-MOF was recorded. The experimental results showed that the luminescence intensity of the Zn-MOF was not affected much after four recycling cycles (Appendix A). This indicated that Zn-MOF could be reused for sample detection after simple solvent washing.

### 2.7. Luminescence Sensing of the Zn-MOF for TC in Urine and Aquaculture Wastewater Systems

Antibiotics in urine have become an important biomarker for studying human exposure to antibiotics. Therefore, the presence of TC was detected in human urine using the prepared Zn-MOF. The Zn-MOF was immersed in the urine with TC and without TC for three days, which also included some interfering agents (NaCl, hippuric acid, creatinine, glucose, urea, L-cysteine, etc.). The results of the experiments are shown in Figure 8a. It was found that only the presence of TC in urine quenches the luminescence of the Zn-MOF, and the quenching effect of TC on the luminescence of the Zn-MOF was not affected by the interfering agents, confirming that the Zn-MOF has high selectivity for sensing TC. Elevated levels of TC in water bodies due to the large-scale use of tetracycline antibiotics in the pharmaceutical and aquaculture industries pose an increased ecological risk to the environment. Therefore, it is necessary to detect TC in aquaculture wastewater plants. TC was added to a suspension of the Zn-MOF containing some inorganic material and additional antibiotics (sulfadiazine (SDZ) and sulfamethazine (SMZ)). It was found that only the presence of TC in aquaculture wastewater quenches the luminescence of the Zn-MOF, and the quenching effect of TC on the luminescence of the Zn-MOF was not affected by the interfering agents (Figure 8b). It was observed that the Zn-MOF had high selectivity for the detection of TC. These findings suggest that the prepared Zn-MOF has the potential to detect TC for practical applications.

### 2.8. The Luminescence Sensing Mechanism

The luminescence sensing mechanism of the Zn-MOF was investigated. Firstly, the PXRD pattern of the Zn-MOF immersed in TC and acetone solution for 48 h was obtained. It can be seen from Figure 9a that the peak positions and intensities of the spectra, before and after the detection of TC and acetone solutions, remained unchanged. This indicates that the structure of the Zn-MOF remained intact after the luminescence experiment. This implies that the prepared Zn-MOF was stable in TC and acetone solutions, and the luminescence quenching of the Zn-MOF by TC and acetone cannot be attributed to the collapse of the framework.

The targeted substrates for detection, TC, and acetone were studied using UV–Vis absorption. Figure 9b shows that TC had a broad absorption band in the range of 300–400 nm and acetone had a strong absorption band in the field of 200–300 nm. The Zn-MOF was observed to have a strong emission band at 385 nm and an excitation band at 305 nm. Therefore, the UV absorption spectra of TC and acetone overlapped with the excitation spectra of the Zn-MOF complex, indicating that the luminescence quenching of the Zn-MOF by TC and acetone was caused by the fluorescence resonance energy transfer (FRET). This suggests that the absorption of incident light by TC and acetone was competitive with that of the prepared Zn-MOF [47,48].

Similarly, photoinduced electron transfer (PET) was also considered as a possible luminescence quenching mechanism. DFT calculations were also performed to investigate the detection mechanism of the Zn-MOF for TC and acetone (Figure 10). According to DFT, the LUMO of H_3_L (−1.17 eV) and the LUMO of BMP (−1.96 eV) were higher than the LUMO of TC (−2.31 eV) but lower than that of acetone (−0.37 eV). Since the LUMO level of the TC was in a lower energy state, the properties of H_3_L and BMP ligands mentioned above enabled the transfer of excited electrons from the framework of the Zn-MOF to TC (LUMO energy from −1.17 eV to −2.31 eV). Therefore, according to these findings, both FRET and PET processes can be considered as the reasonable quenching mechanisms.

## 3. Materials and Methods

### 3.1. Materials

Every reagent was bought from a store and utilized straight away. Using an elemental Vario EL analyzer, elemental analyses (C, H, and N) were performed. The KBr pellet method was used to record infrared (IR) spectra using a Bruker Tensor37 spectrophotometer. On a PANalytical X’pert PRO MPD diffractometer using CuKα radiation (λ = 1.5406 Å), experimental powder X-ray diffraction (PXRD) was performed. Using a nitrogen environment and a heating rate of min^−1^ from room temperature to 800 °C, thermogravimetric analysis (TGA) was performed using an HCT-2 thermal analyzer. An FL7000 fluorescence spectrophotometer was used to record the solid and liquid fluorescence spectra at room temperature. Using a U-3900H spectrophotometer, UV–Vis spectroscopy was carried out.

### 3.2. Synthesis of the Zn-MOF

A 25 mL Teflon cup containing Zn(Ac)_2_ (0.2 mmol), H_3_L (0.1 mmol), BMP (0.2 mmol), NaOH (0.4 mL, 1 mol·L^−1^), and water (10 mL) was heated at 120 °C for 72 h, then cooled to room temperature. For C_52,_ H_44,_ Zn_3_ N_10_O_20_, elemental analysis (%) C, 47.09; N, 10.57; H, 3.32; found (%) C, 47.12; N, 10.61; H, 3.34. IR(KBr pellet, cm^−1^), 3590 (w), 3449 (w), 3246 (m), 3131 (s), 1612 (s), 1562 (m), 1522 (m), 1379 (s), 1314 (w), 1256 (m), 1235 (s), 1155 (m), 1075 (s), 1012 (s), 964 (w), 948 (s), 847 (m), 814 (w), 789(w), 764 (s), 737 (s), 691 (m), 648 (m), 517 (w), 452 (w). The CCDC Number of Zn-MOF was detailed in the Appendix B.

### 3.3. X-ray Crystallographic Study

The Cu Kα (λ = 0.154184 nm) was used in an X-ray diffractometer to gather the crystal data. The SHELXL 97 and SHELXL 97 programs were used to solve the structure. The coordinates of the hydrogen atoms were established using the theoretical hydrogenation approach and were then adjusted using the full-matrix least-squares method. The crystallographic data are summarized in Appendix A, and the selected bond lengths and angles are presented in Appendix A.

### 3.4. Luminescence Measurements

The Zn-MOF powder sample (3 mg) was added to deionized water (3 mL); then, the solution was ultrasonicated for 30 min to form a uniform suspension. The suspension was added to a quartz cuvette. The luminescence emission spectra were recorded at the excitation of 305 nm after each incremental addition of 10 μL TC solution (2 × 10^−4^ mol/L) at room temperature.

The quenching constant was calculated using the Stern-Volmer (SV) equation:

I_0_/I−1 = K_SV_ [A]. Here, K_SV_ is the quenching constant; [A] is the molar concentration of TC; and I_0_ and I are the luminescence before and after TC addition strength.

The LOD of the Zn-MOF for TC can be determined as follows:

LOD = 3σ/K_SV_. Here, LOD is the limit of detection, and σ is the standard deviation of the three repeated luminescence measurements of the Zn-MOF in a blank aqueous solution.

## 4. Conclusions

A new Zn-MOF, [Zn_3_(BMP)_2_L_2_(H_2_O)_4_]·2H_2_O, was designed and synthesized using BMP and H_3_L ligands. The synthesized Zn-MOF had a three-dimensional framework in which the BMP ligand adopted the μ_3_:η^1^η^1^η^1^ coordination mode, while the L ligand adopted the μ_1_:η^1^η^0^/μ_1_:η^1^η^0^/μ_0_:η^0^η^0^ coordination mode. The Zn-MOF exhibited excellent luminescence in solids and solutions, and the luminescence properties remained stable at pH values ranging from 4 to 10. The Zn-MOF fabricated in this study can be used as a luminescent sensor to detect acetone and TC in water. The detection limits of the Zn-MOF were found to be 3.34 µM and 0.1597% for TC and acetone, respectively. TC could be detected by the Zn-MOF in urine and aquaculture wastewater systems. The luminescence quenching mechanisms of the Zn-MOF were investigated in detail using experimental methods and theoretical calculations. These findings suggest that the Zn-MOF is an excellent luminescent material that can be used for detecting environmental pollutants such as TC and acetone.

## Data Availability

Not applicable.

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
