# Peer review of "A Zn(II)–Metal–Organic Framework Based on 4-(4-Carboxy phenoxy) Phthalate Acid as Luminescent Sensor for Detection of Acetone and Tetracycline"

_molecules, 2023, doi:10.3390/molecules28030999_

Round 1

Reviewer 1 Report

In this paper, a new Zn MOF material was synthesized. This material can achieve high sensitivity detection of acetone under different acids and bases and tetracycline under different working conditions. The idea is clear and the demonstration is relatively substantial. However, some minor problems still should be corrected:

1. It recommends that the fluorescence spectrogram range of Figure 1 maybe keep consistent.

2. The interpretation in Figure 3 shows that “under strongly acidic conditions (pH < 3), H3L ligand may not have coordinated well with metal Zn2+ ions”. Is there any ICP test for Zn content or other data as evidence? XRD only indicates that the pH value is stable at 4. What about pH values at 3 or 2? The fluorescence of pH from 5 to 2 has been showing a gradual downward trend.

3. It was previously considered that the material was stable at pH 4-12. Why did the following TC test show that the material was stable within the range of 4-10, Refer shows that "the optimum ph range for tetracyclines complexing with metal ions is weak acidity (pH 2-6)". If the test range in question 2 can be extended to the position with stronger acidity, does the material have similar properties? Or does the complexation of Zn metal ions affect the properties of the material.

4. Drawing specification, it is recommended to check the full text one by one. For example, the legend size of horizontal and vertical coordinates not the same in Figure 2 and the ”ex“ should be subscript, and there is no legend of horizontal coordinates in Figure 3. The icon in Figure 4 suggests careful inspection and correction of the row spacing and other problems.

5. Figure 10 should be "LUMO" not "LOMO".

Author Response

Response to Reviewer 1 Comments

Point 1: It recommends that the fluorescence spectrogram range of Figure 1 maybe keep consistent.

Response 1: The fluorescence spectrogram range of Figure 2 has been adjusted to be consistent.

Point 2: The interpretation in Figure 3 shows that “under strongly acidic conditions (pH < 3), H3L ligand may not have coordinated well with metal Zn2+ ions”. Is there any ICP test for Zn content or other data as evidence? XRD only indicates that the pH value is stable at 4. What about pH values at 3 or 2? The fluorescence of pH from 5 to 2 has been showing a gradual downward trend.

Response 2: XRD indicates that the Zn-MOF is stable at pH 4-12. In our previous experiments, when the Zn-MOF sample was immersed in aqueous solutions with pH=3 or 2, the material was almost completely dissolved, the sample could not be collected and XRD could not be measured. The fluorescence of the solution at pH from 5 to 2 has been showing a gradual downward trend. The experimental results indicated that the Zn-MOF skeleton was collapsed and H3L ligand may not have coordinated well with metal Zn2+ ions under strongly acidic conditions (pH < 3).

Point 3: It was previously considered that the material was stable at pH 4-12. Why did the following TC test show that the material was stable within the range of 4-10, Refer shows that "the optimum ph range for tetracyclines complexing with metal ions is weak acidity (pH 2-6)". If the test range in question 2 can be extended to the position with stronger acidity, does the material have similar properties? Or does the complexation of Zn metal ions affect the properties of the material.

Response 3: XRD and fluorescence experiments show that Zn-MOF is stable in aqueous solutions in the range of pH 4-12. At first, we tested the sensing performance of Zn-MOF to detect TC in a water environment, and the result shows that Zn-MOF has a good sensing ability to detect TC. In order to expand the range of detection, then we selected acidic and alkaline conditions to carry out the fluorescence sensing experiment on TC. Taking pH=4 or 10 as an example, and the sensing ability of the Zn-MOF to TC was also tested. The experimental results show that the Zn-MOF has good sensing ability on TC  in aqueous solution with relatively wide pH value range. The Zn-MOF material is dissolved and unstable in a strongly acidic solution (pH < 3), this material cannot be used as sensor for TC. The complexation of Zn metal ions with the ligands forms new 3D Zn-MOF architecture. Compared with ligands, the Zn-MOF has a stable structure and good luminescence properties does affect the properties of the material, which is conducive to sensing applications.

Point 4: Drawing specification, it is recommended to check the full text one by one. For example, the legend size of horizontal and vertical coordinates not the same in Figure 2 and the ”ex“ should be subscript, and there is no legend of horizontal coordinates in Figure 3. The icon in Figure 4 suggests careful inspection and correction of the row spacing and other problems.

Response 4: The full text has been checked, and we have adjusted the wrong pictures and other problems. The legend size of horizontal and vertical coordinates has been modified to the same in Figure 2, and the ”ex“ was changed to subscript. The legend of horizontal coordinates in Figure 3 has been added. The row spacing issue of Figure 4 has also been fixed.

Point 5: Figure 10 should be "LUMO" not "LOMO".

Response 5: "LOMO" has been modified to "LUMO" in Figure 10.

Reviewer 2 Report

In this paper, Zhao et al. proposed and synthesized new Zn-based MOF base on 4-(4-carboxy phenoxy) phthalate acid and 3,5-bis(1-imidazolyl) pyridine. In addition, the authors report that tetracycline and acetone can effectively quench the luminescence of Zn-MOF in aqueous solution, thus allowing Zn-MOF to be used as a sensor to detect TC and acetone. Overall, this work has performed very well in detailed experimental studies and mechanistic analysis and has important implications for the study of MOFs in photoluminescence. I would like to recommend it for publication in Molecules after the following point can be well addressed.

1. The authors do not provide specific experimental methods for fluorescence testing in the manuscript, which need to be added.

2. The authors should supplement the morphological properties of the Zn-MOF crystal in the manuscript (SEM and optical microscopy).

3. The authors should upload the cif file of the resolved Zn-MOF crystal structure to CCDC and provide the download link in the manuscript.

4. Reference selection is good in the manuscript. It is recommended to add recent articles on design strategies and applications of MOF materials. (e.g., 10.1021/jacs.1c11750, 10.1002/anie.202105830).

5. Authors should double-check their manuscripts before submitting a revision, there are several expression errors. E.g. “Table S1. Coordination complyer"; “Table S2.  Symmetrytransformations”

Author Response

Point 1: The authors do not provide specific experimental methods for fluorescence testing in the manuscript, which need to be added.

Response 1: The specific experimental methods for fluorescence testing have been added in the manuscript (3.4. Luminescence measurements). The Zn-MOF powder sample (3 mg) was added to deionized water (3 mL), then the solution was ultrasonicated for 30 min to form a uniform suspension. The suspension was added to a quartz cuvette. The luminescence emission spectra were recorded at the excitation of 305 nm after each incremental addition of 10 μL TC solution (2×10-4 mol/L) at room temperature.

Point 2:The authors should supplement the morphological properties of the Zn-MOF crystal in the manuscript (SEM and optical microscopy).

Response 2: The morphological properties of the Zn-MOF crystal have been added in 2.2. Morphological properties and thermal stability of Zn-MOF. The morphologies of the Zn-MOF were described through optical microscopy and SEM. The Zn-MOF was colorless and transparent blocky crystals and featured a 3D structure with a rough surface (Figure S1).

Point 3: The authors should upload the cif file of the resolved Zn-MOF crystal structure to CCDC and provide the download link in the manuscript.

Response 3: The cif file of the resolved Zn-MOF crystal structure has been uploaded to CCDC and provided the download link in the manuscript(Appendix A). CCDC 2167668 contains supplementary crystallographic data and can be obtained free of charge via www.ccdc.cam.ac.uk/structures/.

Point 4: Reference selection is good in the manuscript. It is recommended to add recent articles on design strategies and applications of MOF materials. (e.g., 10.1021/jacs.1c11750, 10.1002/anie.202105830).

Response 4: The recent articles on design strategies and applications of MOF materials (e.g., 10.1021/jacs.1c11750, 10.1002/anie.202105830) were cited in the references [17,18].

Point 5: Authors should double-check their manuscripts before submitting a revision, there are several expression errors. E.g. “Table S1. Coordination complyer"; “Table S2.  Symmetrytransformations”

Response 5: The full text has been checked, and we have adjusted these problems. For example, "Coordination complyer" has been modified to "complex" in Table S1. And "Symmetrytransformations" has been adjusted to "Symmetry transformations" in Table S2.

Reviewer 3 Report

This manuscript by Wang et al. reports a Zn(â…¡)-metal-organic framework base on 4-(4-carboxy phenoxy) phthalate acid as luminescent sensor for detection of acetone and tetracycline. I support its publication after minor revisions.

1. I suggest authors to add a scheme describing the main content of this study at the beginning of the manuscript.

2. There are some errors in the figures, for example, the axe label in Figure 3b and "LOMO" in Figure 10.

3. More background about the sensors for acetone and tetracycline should be added in INTRODUCTION.

4. Please provide the details for LOD calculation.

Author Response

Response to Reviewer 3  Comments

Point 1: I suggest authors to add a scheme describing the main content of this study at the beginning of the manuscript.

Response 1: The scheme describing the main content of this study has been added (Scheme 1).

Point 2:There are some errors in the figures, for example, the axe label in Figure 3b and "LOMO" in Figure 10.

Response 2: The full text has been checked, and we have adjusted these problems. "LOMO" has been modified to "LUMO" in Figure 10. The axe label in Figure 3 has been added.

Point 3: More background about the sensors for acetone and tetracycline should be added in INTRODUCTION.

Response 3: Related research on the sensors for acetone and tetracycline was added in INTRODUCTION. Currently, high-performance liquid chromatography (HPLC), capillary electrophoresis (CE), and immunoassay are common methods for detecting environmental pollutants, such as tetracycline and acetone. However, these methods generally require expensive and sophisticated instruments, complex pre-processing procedures, and skilled technicians, which are not conducive to rapid and routine monitoring. Therefore, it is crucial to develop a fast and simple method for detecting TC and acetone.

In recent years, MOFs have also made positive advances in detecting tetracycline and acetone. Huang et al. synthesized a Fe-modified MXene-derived MOF that can be used as a high-performance acetone sensor. Gan et al. proposed fluorescent Eu-MOF designed to detect TC rapidly in food samples. The sentence was added in INTRODUCTION.

Point 4:  Please provide the details for LOD calculation.

Response 4: The details for the LOD calculation method have been added in 3.4. Luminescence measurements.

The quenching constant was calculated by Stern-Volmer (SV) equation: I0/I-1 = KSV [A]. Here, KSV is the quenching constant; [A] is the molar concentration of TC; I0 and I are the luminescence before and after TC addition strength.

The LOD of Zn-MOF for TC can be determined as follows: LOD = 3σ/KSV. Here, LOD is the limit of detection, σ is the standard deviation of the three repeated luminescence measurements of Zn-MOF in a blank aqueous solution.

Round 2

Reviewer 2 Report

My comments were addressed satisfactorily. The manuscript is much improved, and I'd be happy to see it published.